# Experimental Study on Machining Engineering Ceramics by Electrochemical Discharge Compound Grinding

**DOI:** 10.3390/ma12162514

**Published:** 2019-08-07

**Authors:** Kun Xu, Zhaoyang Zhang, Jingbo Yang, Hao Zhu, Xiaolong Fang

**Affiliations:** 1School of Mechanical and Engineering, Jiangsu University, Zhenjiang 212013, China; 2Jiangsu Key Laboratory of Precision and Micro-Manufacturing Technology, Nanjing 210016, China

**Keywords:** engineering ceramics, electrochemical discharge, mechanical grinding, micro-groove machining, surface integrity

## Abstract

Since engineering ceramics have many characteristics, including hardness, brittleness, and high melting point, traditional machining methods can no longer play a useful role in precision machining. Based on this situation, a platform of electrochemical discharge compound mechanical grinding was constructed and is presented in this paper, and machining experiments of micro-grooves were carried out in alumina ceramics. Grooves were observed by scanning electron microscope (SEM), and the morphology and the groove width of micro-grooves under different machining parameters were compared and analyzed. Furthermore, in order to study the improvement effect of mechanical grinding on machining quality, the surface roughness of micro-grooves was measured by a confocal material microscope. The results show that as the pulse power supply voltage increases or the frequency decreases, the width of the micro-grooves increases, and the morphology of the micro-grooves first improves and then deteriorates. With the increase of tool electrode rotation speed, the width of micro-grooves first increases and then remains unchanged, and the morphology and the surface roughness of micro-grooves first improves and then remains stable. Finally, the optimal parameters (power voltage of 20 V, pulse frequency of 400 Hz, electrode rotation speed of 600 rpm) were chosen to machine micro-grooves with good quality.

## 1. Introduction

Due to their many excellent features, including hardness, high strength, durability, corrosion resistance, high temperature resistance, and low friction [1,2], advanced engineering ceramics have been widely used in the electronics, aerospace, and chemical industries, amongst others [3,4,5,6,7]. However, because of their hardness and brittleness, engineering ceramics are one of the most difficult-to-machine materials, especially in fabrication of components requiring high precision and complex morphology, which seriously limits their popularization and application.

Complementary micro-manufacturing technologies have been introduced in engineering ceramics machining, including lapping, mechanical grinding, abrasive slurry jet micro-machining (ASJM), rotary ultrasonic machining, laser machining, electric discharge machining [8,9] and electrochemical discharge machining [10,11]. Lapping and grinding with planetary kinematics provide an ultra-smooth surface, but the material removal rate is very low [12,13]. Grinding is the most efficient and cost-effective technique in ceramic component machining, using different grinding processes such as high-speed grinding [14] and rotary ultrasonic grinding machining [15,16]. However, such methods inevitably result in damage such as micro-cracks. ASJM has the potential to be a relatively inexpensive technology for machining ceramics, but there is still a significant amount work required for its implementation [17,18]. In addition, rotary ultrasonic machining has the potential to achieve high material removal rates and result in relatively low surface damage and strength degradation. However, edge chipping not only compromises geometric accuracy but also possibly causes an increase in machining cost [19,20].

In this paper, electrochemical discharge compound grinding was adopted in Al_2_O_3_ engineering ceramic machining using a grinding needle with diamond grit, with the aim of fabricating structures with high precision and a damage-free surface. In the experimental investigation, micro-grooves were fabricated, and the influence of voltage and frequency of pulse power supply on the morphology and width of micro-groves was determined. Moreover, the effect of mechanical grinding on surface morphology was investigated by changing the revolving speed of the tool.

## 2. Materials and Methods

### 2.1. Principles

As shown in Figure 1a, the tool electrode and the auxiliary electrode were connected to the negative and positive pole of the power supply, respectively, and the electrodes were immersed in the electrolyte. In the pulse duration, hydrogen bubbles were generated around the tool electrode due to electrolytic reaction. Then, a gas layer was formed while the bubbles accumulated on the tool surface, as shown in Figure 1b. Due to the gas layer, the tool electrode was separated from the electrolyte, and the electric potential difference between them increased gradually. When the electric potential difference was large enough, the gas layer broke down and spark discharge occurred. Spark discharge resulted in plasma with a high temperature and high pressure, which struck the workpiece, causing material to melt and allowing it to be removed. In addition, material nearby was intenerated, and the softened material easily removed by grinding.

As shown in Figure 2, there is the pulse period T:T = *t*_duration_ + *t*_off_ = *t*_1_ + *t*_2_ + *t*_off_,(1)
T = 1/*f*,(2)
where *t*_duration_ is the pulse duration time, *t*_off_ is the pulse interval time, *t*_1_ is the formation time of the gas layer, *t*_2_ is the time of the electric discharge process and *f* is the pulse frequency. Figure 2 shows the current waveform during the machining process with 50% duty cycle. During the time of *t*_1_, the current rapidly rises and a lot of bubbles are produced. As the contact area between the tool electrode and the electrolyte decreases due to the bubbles, the current decreases rapidly until the gas layer is formed. Afterwards, spark discharge begins and the gas layer is damaged, an electrochemical reaction occurs, and bubbles are produced to supplement the gas layer.

A low frequency leads to a long pulse period and pulse duration, so that there is enough time for the formation of the gas layer and the electro-discharge process. A high voltage leads to a high electrochemical reaction speed, and a stable gas layer is formed quickly, so that there is a long time remaining for the spark discharge process. In addition, high voltage leads to intense spark discharge, and more material is removed or intenerated. Mechanical grinding is an efficient material removal process, however, for machining engineering ceramics, tool wear and cracks limit the development of grinding. In electrochemical discharge compound grinding, material is softened by spark discharge before being removed by grinding, so the problems of tool wear and cracking can be mitigated.

### 2.2. Experimental Setup

Figure 3 shows the experimental setup for micro-groove machining built by the author team, which includes a computer, a three-axis stage, electric rotating machinery, an electrolyte slot, a pulse generator, an auxiliary electrode, a tool electrode and a workpiece.

The tool electrode and the workpiece were mounted on the electric rotating machinery and the electrolyte slot, respectively. The tool electrode was attached to the electric rotating machinery and the three-axis motion stage, then the rotary motion and feed movement were controlled by the computer though a motion controller. The pulse generator (i.e., power supply) provided pulses for electrochemical discharge, with its negative and positive poles connected to the tool electrode and the auxiliary electrode, respectively. An oscillograph and a current probe was used to monitor the processing current in real time. The metal tool electrode was 45 mm long and 3 mm in diameter, and diamond grit with a particle size of 150 mesh was adhered to the tip of tool. As one of the most common ceramic materials, alumina ceramic plate was chosen as the workpiece. A graphite block was adopted as the auxiliary electrode.

### 2.3. Experimental Details

The designed experimental setup was used for micro-groove fabrication and electrochemical discharge compound grinding of engineering ceramics. In order to obtain micro-grooves with good quality, experiments were developed, and the influences of key parameters, such as power voltage, power pulse frequency and rotation speed of the tool electrode, on the groove width and surface morphology were studied.

Small pieces of alumina ceramic of 2-mm thickness were used as the workpiece, and the size of the auxiliary electrode was 50 mm × 30 mm × 8 mm. The diameter of the tool tip was 0.55 mm. Sodium hydroxide solution of 30 wt.% was chosen as the electrolyte for the electrochemical reaction. Figure 4 shows the processing path of grooves: the machining of the micro-groove was carried out four times, each with a feed path length of 0.5 mm and feed rate of 5 μm /s. The tool returned to the origin point after finishing each feed movement, fed 50 μm in the direction of the micro-groove depth, and then continued to grind along the last feed path. The grooves’ width and surface morphology were observed and measured by a scanning electron microscope (S-3400N, Hitachi, Tokyo, Japan) and a laser confocal microscope (TCS SP5 II, Leica, Wetzlar, Germany).

## 3. Results and Discussions

### 3.1. Effects of Power Pulse Frequency

According to the analysis results in Section 2.1, the power pulse frequency has a significant influence on the electrochemical discharge process. In this section, the following parameters were used for the groove experiments: 50% duty cycle for the power pulses, 20 V for the power voltage and 500 rpm for the rotation speed. Different power pulse frequencies of 200 Hz, 400 Hz, 600 Hz, 800 Hz, 1000 Hz and 2000 Hz were adopted to investigate the effect of pulse frequency on the groove width and surface morphology.

The effect of the pulse frequency on the groove width is shown in Figure 5. When the frequency increased from 200 Hz to 1000 Hz, the groove width decreased rapidly. A high frequency meant a short pulse period, so that the time for the spark discharge process was also short, and less material was removed or intenerated. When the frequency increased from 1000 Hz to 2000 Hz, the groove width decreased slowly. This is because the frequency was too high and there was not enough time for the formation of a stable gas layer, and the spark discharge would not remove or intenerate enough material.

Figure 6 shows the surface morphology of grooves produced with frequencies of 200 Hz,400 Hz and 1000 Hz. It is obvious that the groove machined with the frequency of 400 Hz had the best morphology. When the frequency was 200 Hz, the time for the spark discharge process was too long, and the edge of the groove was damaged. When the frequency was 1000 Hz, the time for the electrochemical discharge was too short, and the rate of material removed or intenerated by spark discharge was slow. As a result, more unsoftened material was removed by the grinding process, causing a bad morphology. Therefore, a frequency of 400–600 Hz was considered optimal for the power pulse.

### 3.2. Effects of Power Voltage

According to the analysis results in Section 2.1, the power voltage significantly influenced the material removal rate. In this section, the following parameters were used for the groove machining experiment: 50% duty cycle for the power pulses, 500 Hz for the power pulse frequency and 500 rpm for the rotation speed. Different power voltages of 17 V, 18 V, 19 V, 20 V, 21 V and 22 V were used to analyze the effect of pulse frequency on groove width and surface morphology.

The influence of power voltage on groove width is shown in Figure 7: the groove width increases as the power voltage increases. A high voltage leads to an intense discharge process and a long discharge time in a pulse period. Thus, more material was removed directly and, in addition, more material was softened and able to be removed by mechanical grinding, and a wide groove was then obtained.

Figure 8 shows three photographs of grooves produced with different voltages. When the voltage was 17 V, the spark discharge energy was too small to remove enough material or soften enough material for mechanical grinding, so the material removal rate was too low to fabricate a micro-groove with a good condition, as shown in Figure 8a. A micro-groove with good morphology was obtained with a voltage of 20 V as shown in Figure 8b. As shown in Figure 8c, when the voltage was 22 V, the higher spark discharge energy would lead to defects such as microcracks and corrosion pits. Thus, a voltage of 19–21 V is considered to be appropriate for micro-groove fabrication.

### 3.3. Effects of Rotation Speed

As material is removed by spark discharge and mechanical grinding, the rotation speed of the tool has a significant influence on the width of the groove and the surface morphology. As shown in Figure 9, the groove width increased from 547 μm to 621 μm when the rotation speed increased from 100 rpm to 400 rpm. When the feed rate was fixed, a low rotation speed meant more material was removed during a single rotation and with a large grinding force. As a large grinding force caused wear of the tool and shedding of abrasives, the actual radius of the tool was reduced, and a narrow groove was obtained. When the rotation speed increased from 400 rpm to 1000 rpm, the groove width increased from 621 μm to 636 μm. The increase of rotation speed still increased the groove width slowly when the rotation speed was high. As a surface with good quality was obtained with a small grinding force and sharp abrasives, a high rotation speed improved the surface roughness as shown in Figure 10.

Figure 11 shows the photograph of grooves machined with rotation speeds of 200 rpm and 800 rpm. Compared with the high-quality groove processed at the rotation speed of 800 rpm, the groove prepared at the rotation speed of 200 rpm was wider, with poorer surface roughness and morphology. However, the electrolyte splashes easily and the gas layer is damaged when the tool speed is too high, thus, an optimal rotation speed of 600–800 rpm was chosen to fabricate the micro-groove.

### 3.4. Groove Machining with the Optimal Parameters

By using the optimal machining conditions—power voltage of 20 V, frequency of 400 Hz, 50% duty cycle, and a rotation speed of 600 rpm—micro-grooves were fabricated. Figure 12 and Figure 13 show the SEM photograph and 3D photograph of the groove, respectively, and a micro-groove with a smooth surface and good morphology was prepared.

## 4. Conclusions

In this paper, an experimental setup for electrochemical discharge compound grinding was built and micro-grooves with different parameters were prepared. A machining method with high efficiency and crack-free alumina ceramics was promising. The conclusions can be summarized as follows:Low pulse frequency and large voltage led to a stable gas layer and sufficient discharge energy, which are necessary for material removal in electrochemical discharge compound grinding. With the decrease of frequency or increase of voltage, the groove width increased. However, the surface morphology was damaged when the frequency was too low or the voltage too large. Thus, a frequency of 400–600 Hz and a voltage of 19–21 V are considered to be appropriate.High rotation speed led to a small grinding force and little tool wear, which is good for increasing the material removal rate and improving the surface morphology. However, a too-high rotation speed may cause splashing of the electrolyte and an unstable gas layer. Thus, an optimal rotation speed of 600–800 rpm was chosen.Using the optimal machining parameters of a 20-V voltage, a 400-Hz frequency and a 600-rpm rotation speed, a micro-groove with a smooth surface and good morphology was fabricated by electrochemical discharge compound grinding, and better processing results could be expected in future studies.

## Figures and Tables

**Figure 1 materials-12-02514-f001:**
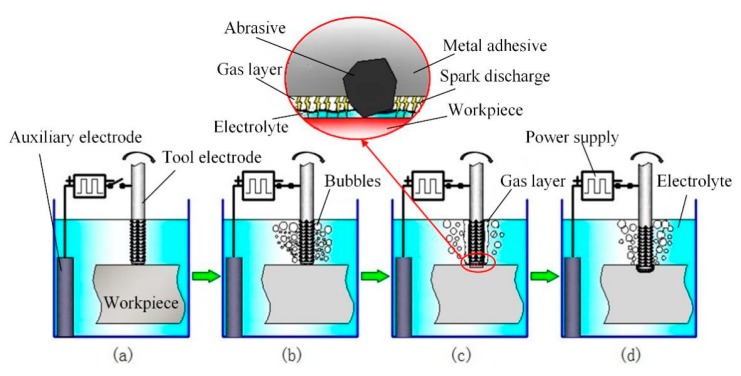
Processing diagram of electrochemical discharge compound grinding: (**a**) raw condition, (**b**) power on and gas layer formation, (**c**) spark discharge occurred, and (**d**) material removed by spark discharge and grinding.

**Figure 2 materials-12-02514-f002:**
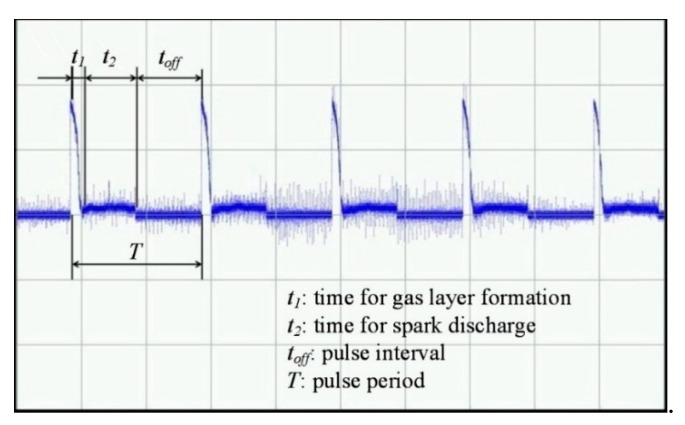
Current waveform of the machining process.

**Figure 3 materials-12-02514-f003:**
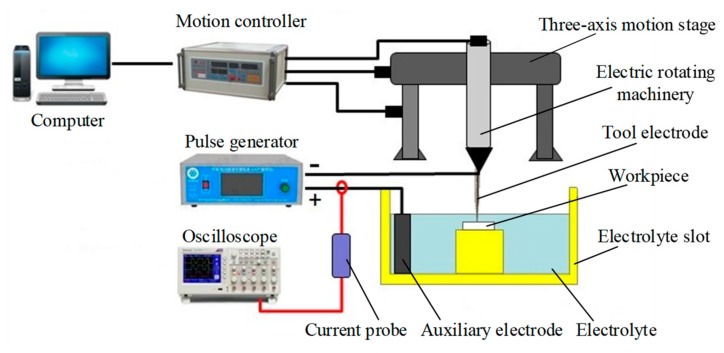
Experimental setup.

**Figure 4 materials-12-02514-f004:**
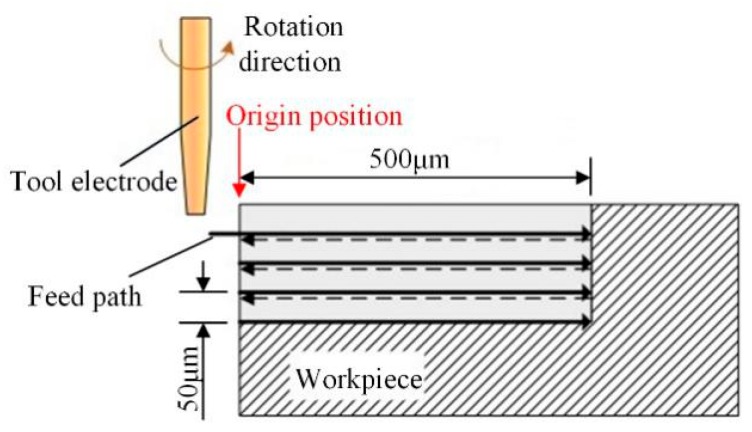
Schematic diagram of the processing path.

**Figure 5 materials-12-02514-f005:**
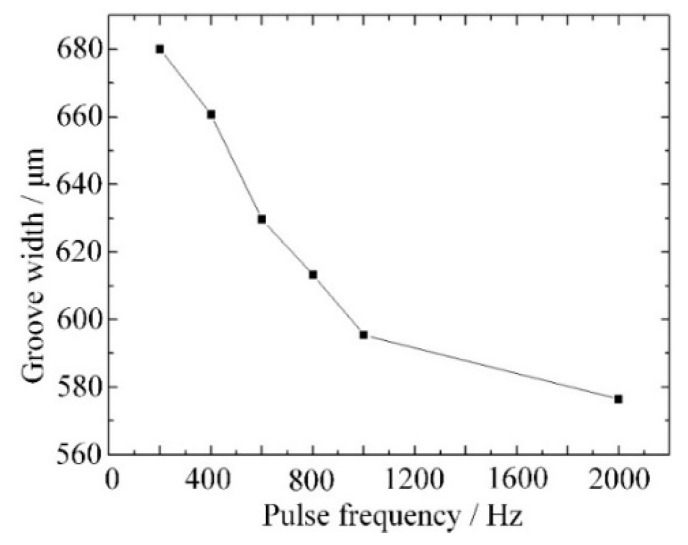
Groove width variation with different pulse frequency.

**Figure 6 materials-12-02514-f006:**
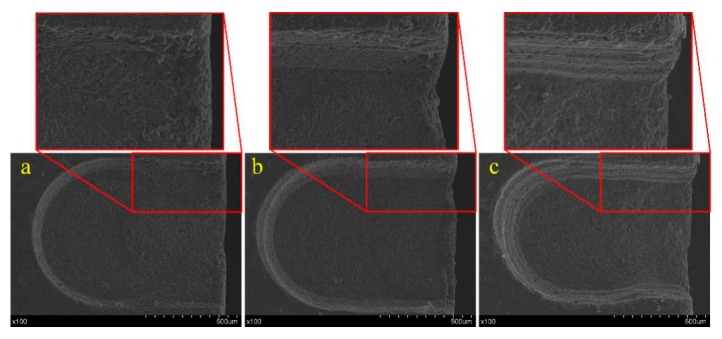
Groove morphology variation with different pulse frequency: (**a**) 200 Hz, (**b**) 400 Hz, and (**c**) 1000 Hz.

**Figure 7 materials-12-02514-f007:**
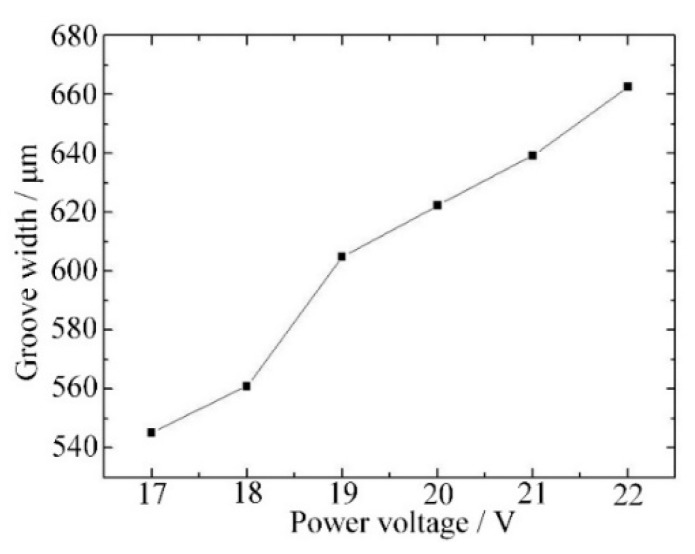
Groove width variation with different power voltage.

**Figure 8 materials-12-02514-f008:**
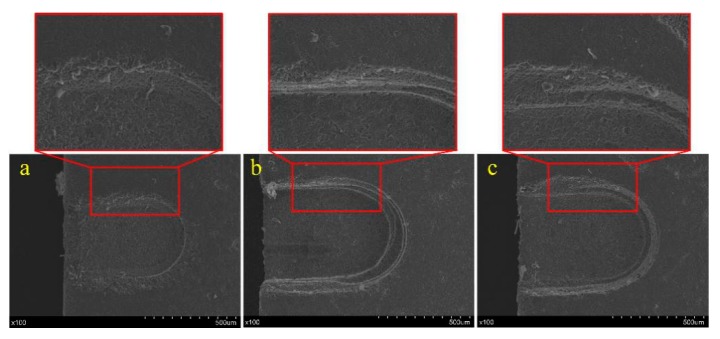
Groove morphology variation with different power voltages: (**a**) 17 V, (**b**) 20 V, and (**c**) 22 V.

**Figure 9 materials-12-02514-f009:**
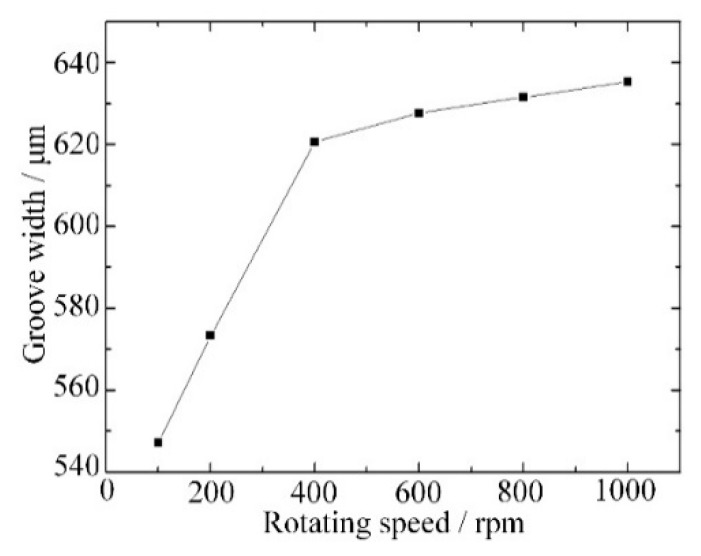
Groove width variation with different rotation speed.

**Figure 10 materials-12-02514-f010:**
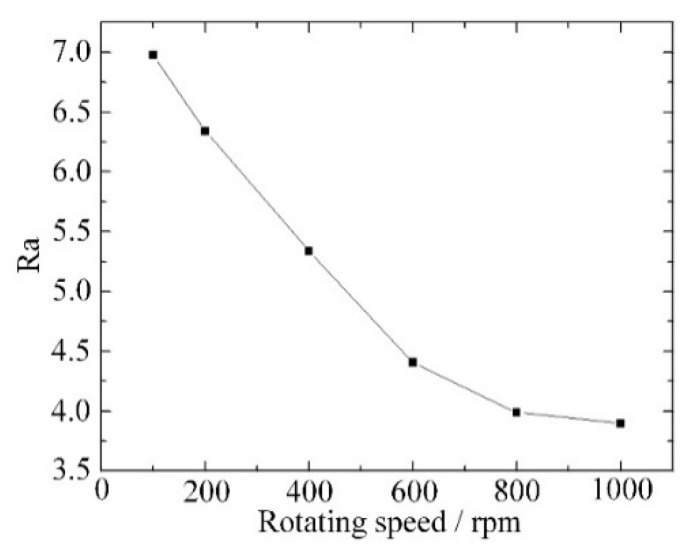
Surface roughness variation with different rotation speed.

**Figure 11 materials-12-02514-f011:**
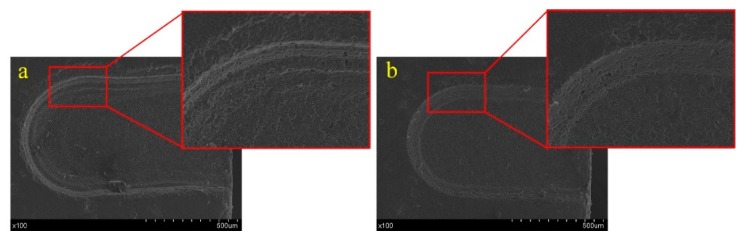
Groove morphology variation with different rotation speed: (**a**) 200 rpm and (**b**) 800 rpm.

**Figure 12 materials-12-02514-f012:**
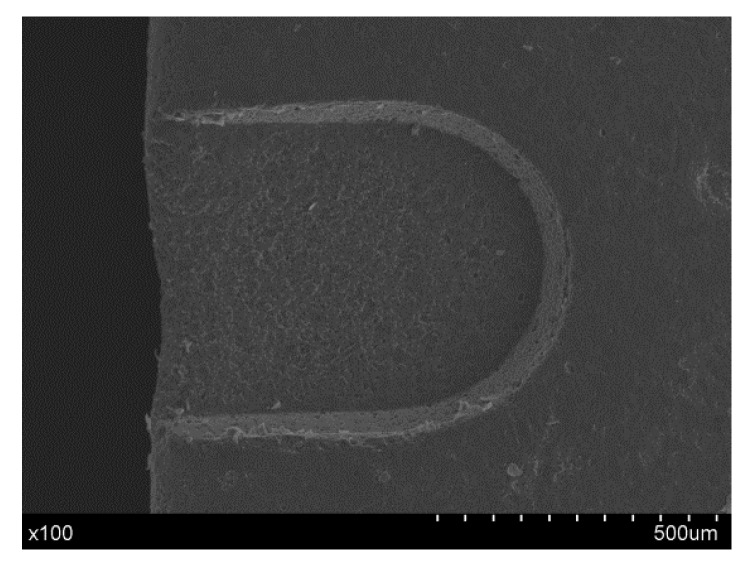
SEM photograph of the groove.

**Figure 13 materials-12-02514-f013:**
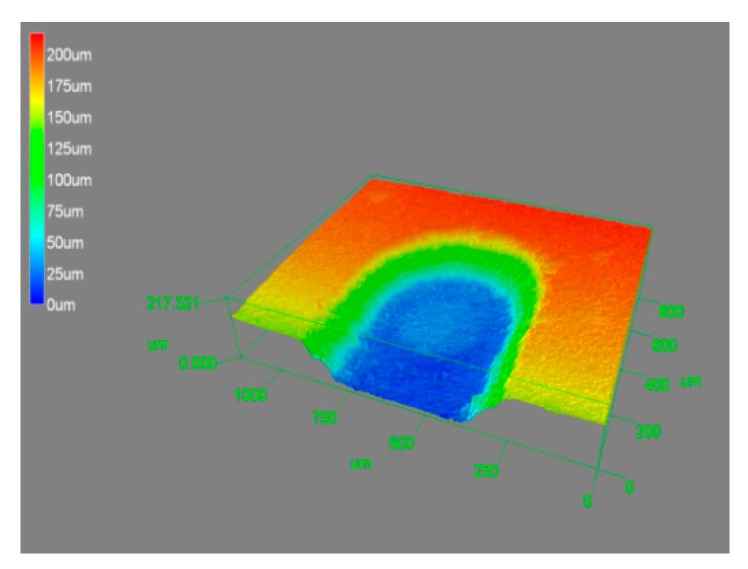
3D photograph of the groove.

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
