# Peer review of "Experimental Study on Machining Engineering Ceramics by Electrochemical Discharge Compound Grinding"

_materials, 2019, doi:10.3390/ma12162514_

Round 1
Reviewer 1 Report
In this paper, experimental setup for electrochemical discharge combined grinding was built and micro grooves with different parameters were prepared. The influence of various parameters such as voltage, power pulse frequency and rotation speed on groove formation mechanism and surface morphology was studied using alumina ceramics as a test object. The optimal parameters of grinding were specified. The reported study is of an engineering nature and gives a good impression, the interpretation of the results and conclusions seem adequate. I recommend it for publication. However, the authors do not highlight the fact that their tests were performed on a specific material (alumina) nor in abstract, nor in conclusions. This accounts for the impression that developed grinding parameters are universal and can be used for any types of ceramics (piezoelectric, ferrite, etc.), but it is unlikely to be so. In this regard, the authors are advised to substantiate why they selected alumina ceramics as a test object. Abstract and conclusions should also include a mention of alumina.
Author Response
Response 1: Following the suggestion of reviewer, the reason of selecting alumina ceramics was added in paragraph 2.2, and mentions of alumina ceramics were added in abstract and conclusions.

Reviewer 2 Report
The small suggestions are included in the uploaded file. The tenses are mixed in the sentences.
The practical aspects should be mentioned in the conclusions. More detailed conclusions should be given.
Other damage free methods for ceramic materials should be mentioned: lapping and grinding with lapping kinematics. Appropriate references:
Wan, L., Dai, P., Li, L., Deng, Z., & Hu, Y. (2019). Investigation on ultra-precision lapping of A-plane and C-plane sapphires. Ceramics International, 45(9), 12106-12112.
Deja, M., List, M., Lichtschlag, L. D., & Uhlmann, E. (2019). Thermal and technological aspects of double face grinding of Al2O3 ceramic materials. Ceramics International https://doi.org/10.1016/j.ceramint.2019.06.206

Author Response
Response 1: Sentences have been corrected following the reviewer’s suggestion.
Response 2: Following the suggestion of reviewer, the conclusions was revised.
Response 3: As Reviewer suggested that sections about the lapping and grinding with planetary kinematics were add to the introduction.
